# The Reverse Side of a Coin: “Factor-Free” Ribosomal Protein Synthesis In Vitro is a Consequence of the In Vivo Proofreading Mechanism

**DOI:** 10.3390/biom9100588

**Published:** 2019-10-08

**Authors:** Alexei V. Finkelstein

**Affiliations:** 1Institute of Protein Research, Russian Academy of Sciences, Pushchino, 142290 Moscow Region, Russia; afinkel@vega.protres.ru; 2Biology Department, Lomonosov Moscow State University, 119192 Moscow, Russia

**Keywords:** ribosome, biosynthesis of polypeptides, elongation factor Tu, EF-Tu, GTP, factor-free process, factor-promoted process, parallel reactions, free energy, anticodon-codon recognition, complementarity, non-complementarity, binding of Aa-tRNA, proofreading, removal of Aa-tRNA

## Abstract

This paper elucidates a close connection between two well-known facts that until now have seemed independent: (i) the quality control (“proofreading”) of the emerging amino acid sequence, occurring during the normal, elongation-factor-dependent ribosomal biosynthesis, which is performed by removing those Aa-tRNAs (aminoacyl tRNAs) whose anticodons are not complementary to the exhibited mRNA codons, and (ii) the in vitro discovered existence of the factor-free ribosomal synthesis of polypeptides. It is shown that a biological role of proofreading is played by a process that is exactly opposite to the step of factor-free binding of Aa-tRNA to the ribosome-exposed mRNA: a factor-free removal of that Aa-tRNA whose anticodon is not complementary to the ribosome-exhibited mRNA codon.

## 1. Introduction

The need for quality control of ribosome-synthesized amino acid sequences was predicted back in the 1950s by Linus Pauling. It is now known that such a control operates according to two key stages [1,2]: (1) control of making only “correct” aminoacyl-tRNA (Aa-tRNA) molecules by “charging” each tRNA with its corresponding amino acid—this stage is considered in numerous publications (see, e.g., [1,3,4,5]) and we will not address it here, and (2) control of the incorporation of only proper amino acid residues into the polypeptide chain growing on the ribosome (Figure 1), that is, of those whose Aa-tRNAs accurately fit to mRNA codons exposed by the ribosome [2,6,7].

The rejection of incorrect Aa-tRNAs occurs in two steps, namely initial selection and proofreading. The former occurs before GTP hydrolysis (performed with the aid of ribosomal elongation factor Tu [2]) and it is based on a much slower binding of incorrect Aa-tRNA than the correct one [8]. This initial kinetic selection will be not addressed here. The latter step, i.e., the proofreading itself, occurs after the irreversible GTP hydrolysis, when the Aa-tRNA is already somehow bound to the ribosome.

In this work, as in a previous short communication [9] (but in more detail), we elucidate the close relationship of the proofreading step of the polypeptide chain quality control with the existence of the in vitro discovered [2,10,11,12,13,14,15] “factor-free” polypeptide synthesis by the ribosome, while the classical in vivo biosynthesis is carried out by the ribosome using elongation factors Tu and G (EF-Tu, EF-G) that hydrolyze GTP molecules [2] (of these two elongation factors, only EF-Tu involved in the Aa-tRNA binding will be of interest for us).

We shall examine the relationship between (i) quality control in biosynthesis and (ii) the existence of two possible pathways of Aa-tRNA binding the ribosome-exhibited mRNA codon: (a) the factor-dependent pathway, in which the high-energy GTP molecule is hydrolyzed with the aid of the elongation factor Tu (i.e., EF-Tu) that stimulates biosynthesis (Figure 2), and (b) the factor-free pathway that is not supported by the additional free energy of GTP hydrolysis (Figure 3).

## 2. Energetics and Kinetics of Aa-tRNA Binding to the Ribosome

Figure 4 shows the “factor-free” elongation cycle of the ribosome in somewhat more detail (described in [2,16,17]) together with generalized plots showing a change in the total system’s free energy and its main constituents during this cycle. It concerns the case of Aa-tRNA complementary to the exposed mRNA codon, but similar plots can be drawn also for non-complementary Aa-tRNA and for factor-promoted, as well as for factor-free cycles.

Exploring proofreading, one should compare those parts of these free energy plots that concern formation and further decay, or transformation of the complex of Aa-tRNAs with “ribosome•mRNA” (Figure 5). Plots of this kind suggest an easy way to compare the rates of different individual transitions, because these rates are proportional to exponents of the heights of the corresponding barriers, while the pre-exponential multipliers play a secondary role [1,5].

Figure 5 reflects the following basic facts: (i) factor-promoted Aa-tRNA binding is faster than it’s factor-free counterpart; (ii) the ribosome•mRNA complex strongly binds the complementary Aa-tRNA and poorly binds the non-complementary Aa-tRNA; (iii) the ribosome loses the loosely bound (i.e., having high free energy) non-complementary Aa-tRNA via a process that is the reverse of the factor-free binding: the barrier on this pathway is lower than on any other pathway from the complex “non-complementary Aa-tRNA•(ribosome•mRNA)”.

## 3. The (Ribosome•mRNA) Complex with Aa-tRNA Can be Initially Formed by the Factor-Promoted Pathway, but, If Unstable, It Then Decays Using A Reverse Move Along the Factor-Free Pathway

It is well known that, for each reaction, there is a reverse one. 

The binding of Aa-tRNA to the ribosome in the presence of a GTP-bearing elongation factor Tu (Figure 2) is practically irreversible, since it is accompanied by GTP hydrolysis (chemical reaction Tu•GTP → Tu•GDP + P_i_) when the standard free energy drops by ≈7 kcal/mol [1] (which is much higher than the energy of a spontaneous thermal fluctuation, *k*_B_*T* ≈ 0.6 kcal/mol at room temperature *T* ≈ 300 ^°^K). This large free energy decrease caused by the direct reaction practically rules out a spontaneous reverse reaction (the factor-dependent decay of the “ribosome•Aa-tRNA” complex) comprising the highly energy-consuming Tu•GDP + P_i_ → Tu•GTP chemical reaction. 

In contrast, the factor-free binding of Aa-tRNA to the ribosome is reversible, since it is not accompanied by a high-energy effect of this kind (Figure 3). Thus, if the ribosome•Aa-tRNA complex is unstable, the use of the reverse movement along the factor-free pathway is a natural way for its decay.

Therefore, in principle, the (ribosome•mRNA)•Aa-tRNA complex is normally formed via the fastest, factor-promoted pathway (which indeed happens in vivo), but decays (if this complex is unstable due to the non-complementarity of the “improper” Aa-tRNA) via the reverse movement along the factor-free one (Figure 5 and Figure 6). This reverse movement is used by proofreading in vivo, however, in the absence of GTP or EF-Tu, the direct movement along the same pathway leads to factor-free protein synthesis.

The decay of the (ribosome•mRNA)•Aa-tRNA complex by the factor-free pathway does not occur when the Aa-tRNA is complementary to the ribosome-exposed mRNA codon, i.e., it sticks to it strongly. However, the binding of the ribosome (and the exposed mRNA codon) to a non-complementary Aa-tRNA is much weaker. Such a non-complementary complex decays, and it can do so only by the factor-free pathway (Figure 5 and Figure 6). This unproductive decay of non-complementary complexes should lead to an enormous decrease in the rate of incorporation into the emerging polypeptide of those amino acids that are not mRNA-encoded, and this leads to a sharp decrease in the number of translation errors. Both of these phenomena are observed experimentally [20,21].

## 4. Discussion and Conclusions

The factor-dependent formation of complexes of Aa-tRNAs with ribosomes (for both complementary and non-complementary Aa-tRNAs) and the subsequent factor-free removal of the non-complementary Aa-tRNAs were noted already by Hopfield [6]. The selectivity of these processes was studied experimentally [7] and by molecular dynamics [22], but the connection between the pathway of removal of the non-complementary Aa-tRNA and a part of the factor-free pathway of polypeptide synthesis was not considered in those studies.

Thus, the factor-free (not requiring GTP hydrolysis) pathway for the attachment of Aa-tRNA to the ribosome plays an important biological role—with its help, proofreading of the arising amino acid sequence is carried out. More specifically, the biological role is played by the process that is the reverse of the factor-free attachment of Aa-tRNA to the ribosome—that is, the removal of the non-complementary Aa-tRNA using the same pathway.

In this case (see Figure 6), a futile (where GTP hydrolysis produces heat only) ribosome-catalyzed cycle may occur. This “idle” cycle consists of the formation and subsequent disintegration of the complex of non-complementary Aa-tRNA with EF-Tu and GTP. Such a futile cycle with a drastically increased GTP consumption was found in [23,24] using non-complementary Aa-tRNA.

Figure 5 and Figure 6 show that the quality control of the growing amino acid can occur (1) due to different rates of binding of the complementary and non-complementary Aa-tRNAs to the ribosome•mRNA complex, and (2) due to further rejection of the non-complementary Aa-tRNAs by this complex. The former process is factor-dependent while the latter is factor-free, which agrees with the results described in [7].

It is noteworthy that, if there were only one factor-promoted pathway for the attachment of Aa-tRNAs to the ribosome, the non-complementary Aa-tRNA could have been removed from this complex only by the same pathway. This would require GTP synthesis from GDP and phosphate, which is practically impossible thermodynamically (at typical cellular concentrations of GTP, GDP and phosphate). Then, the occurrence of non-complementary Aa-tRNA in the ribosome would lead to an irreversible biosynthesis failure. The presence of a factor-free (and thus not involving GTP) pathway of binding (and hence, removal as well) of Aa-tRNA makes it possible to remove the non-complementary (and therefore poorly bound to the ribosome) Aa-tRNA without GTP synthesis (Figure 5 and Figure 6).

Thus, we see that the pathway used in vivo for proofreading, i.e., preventing biosynthesis errors can be used in vitro for the factor-free (more precisely, EF-Tu-free) synthesis of polypeptides.

The essence of this paper is based on two fundamental facts: (i) that there are two pathways of Aa-tRNA attachment to the ribosome (and thus, the reverse reactions will detach Aa-tRNA from the ribosome), and (ii) that one of these pathways is GTP-dependent and is accompanied by a large free energy decrease due to GTP hydrolysis, while another pathway is not accompanied by hydrolysis and the corresponding free energy decrease.

These two facts are enough to conclude that, if the resulting (Aa-tRNA)-ribosome complex is unstable, the Aa-tRNA will detach from the ribosome using the reverse movement along just the GTP-independent pathway, because the usage of reverse movement along the GTP-dependent pathway is prohibited by the necessity of the large free energy increase. 

## Figures and Tables

**Figure 1 biomolecules-09-00588-f001:**
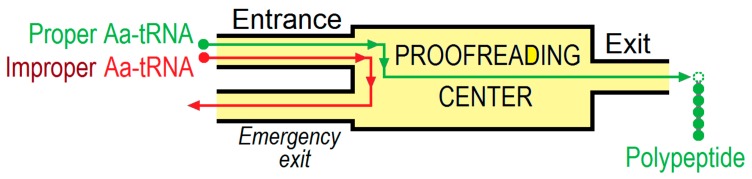
A scheme of proofreading.

**Figure 2 biomolecules-09-00588-f002:**
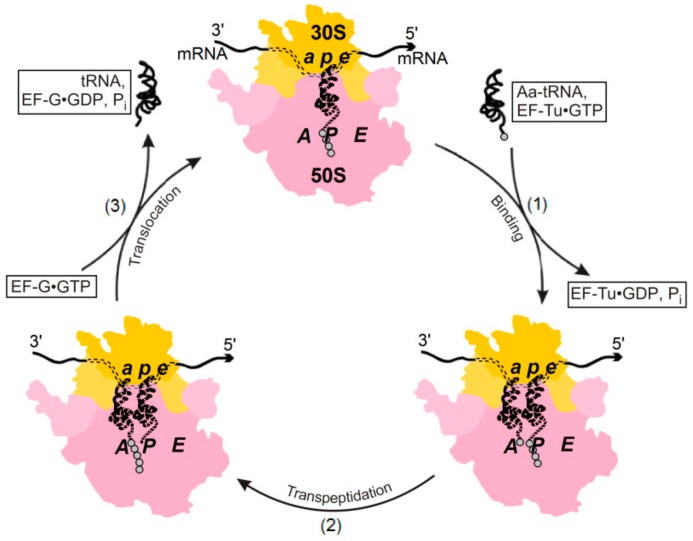
Factor-depending working cycle of the ribosome in vivo. The small (30S in prokaryotes) ribosomal subunit is shown in yellow—it is bound to the mRNA and contains the decoding center (DC) which consists of the ***a***, ***p***, and ***e*** sites that have to bind the anticodon arms of tRNAs, thus forming the (tRNA anticodon)-(mRNA codon) duplexes at different stages of the working cycle. The large (50S in prokaryotes) ribosomal subunit is shown in pink—it contains the peptidyl transferase center (PTC), which consists of the ***A***, ***P***, and ***E*** sites that have to bind the acceptor arms of tRNAs at different stages of the working cycle. Small gray circles bound to tRNAs denote amino acid residues. Each of the three shown stages is irreversible, because they include an exergonic chemical reaction: at stage (2), transpeptidation, and at stages (1) and (3), the auxiliary elongation factors EF-Tu and EF-G cleave the GTP molecules. Adapted from [2,16,17] with minor modifications.

**Figure 3 biomolecules-09-00588-f003:**
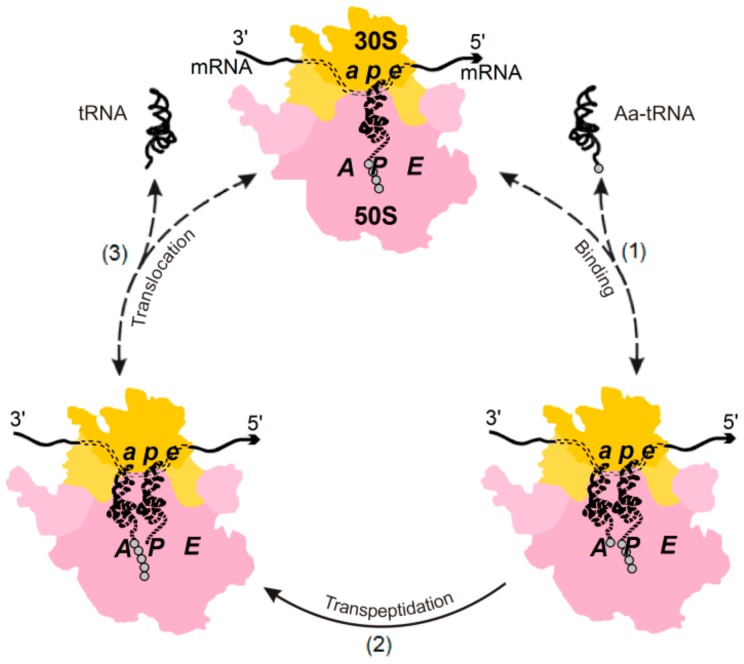
The elongation cycle of the translating ribosome in the simplest “factor-free” in vitro process. The only difference between Figure 3 and Figure 2 is the absence (in Figure 3) of the elongation factors EF-Tu and EF-G together with the GTP molecules. Because of that, stages (1) and (3), which now do not include exergonic chemical reactions, can be reversible (which is shown by dashed two-way arrows), while stage (2), transpeptidation, remains irreversible. Adapted from [2,16,17] with minor modifications.

**Figure 4 biomolecules-09-00588-f004:**
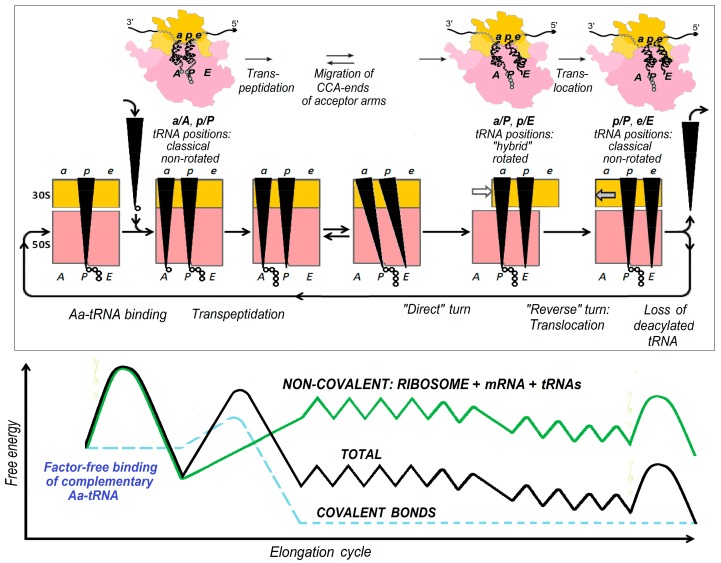
Top panel: the entire elongation cycle of the translating ribosome for the “factor-free” process. All the shown events, except for transpeptidation, require significant movements of the ribosome parts and/or mRNA and tRNAs. The ribosome is in the open (unlocked) state when it turns, when the deacylated tRNA is leaving, and when the Aa-tRNA is binding—in other states, it is in the closed (locked) state [2,18,19]. Bottom panel: generalized plots showing the free energy change during one “factor-free” elongation cycle (for the case when the anticodon of Aa-tRNA is complementary to the exposed mRNA codon) [17]: the covalent bond energy (blue dashed line), non-covalent interactions within the (ribosome•mRNA)•tRNAs complex (green line), and the sum of these two free energies (black line). The first large free energy barrier corresponds to the transition state at the stage of Aa-tRNA binding, and the last, to that at the stage of tRNA unbinding, and the central one to the activation barrier at the stage of transpeptidation (chemical reaction that is an energy source for the entire factor-free elongation cycle). The growth of free energy of the non-covalent interactions within the (ribosome•mRNA)•tRNAs complex occurring at the latter stage is due to poor binding of the acceptor arms of the newly formed peptidyl-tRNA to the A site and the deacylated tRNA to the P site. The small bumps on the free energy profile at the stages of forward and reverse turns symbolize free-energy barriers for the diffusion motion of the 30S subunit relative to the 50S one. Adapted from [17] with some modifications.

**Figure 5 biomolecules-09-00588-f005:**
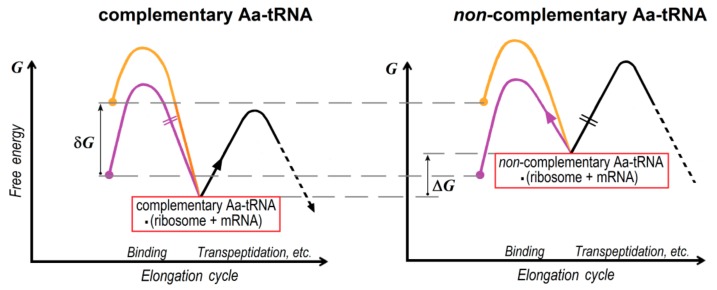
Scheme of the total free energy change in formation and further transformation of the Aa-tRNA•(ribosome•mRNA) complex for the complementary and non-complementary Aa-tRNAs. Purple line: factor-free binding; orange line: EF-Tu•GTP-promoted binding. Δ*G* (>>*k*_B_*T*): the free energy difference between the binding of non-complementary and complementary Aa-tRNA to the ribosome•mRNA complex. δ*G* (>>*k*_B_*T*): the free energy difference between the factor-free and Tu factor-promoted binding of Aa-tRNA to the ribosome•mRNA complex, i.e., the free energy outcome of the exergonic reaction of GTP hydrolysis, EF-Tu•GTP → EF-Tu•GDP + P_i_. The time of a reaction is exponentially dependent on the barrier height. The main, i.e., the fastest (overcoming the lower free energy barrier) pathway of transformation of the Aa-tRNA•(ribosome•mRNA) complex is shown by an arrow, and the slower pathway is crossed out.

**Figure 6 biomolecules-09-00588-f006:**
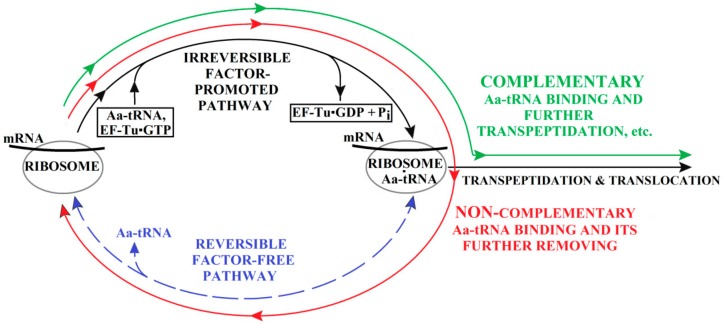
Proofreading: Rejection of the amino acid brought by the non-complementary Aa-tRNA (red solid arrow), but incorporation into the growing polypeptide chain of the amino acid brought by the Aa-tRNA complementary to the ribosome-exhibited mRNA codon (green solid arrow). The center of the scheme shows two parallel pathways for penetration into the ribosome of the “new” amino acid brought by the Aa-tRNA. The irreversible factor-promoted pathway (black one-way arrow) is supported, while the reversible factor-free pathway (blue dashed two-way arrow) is not supported by the free energy of GTP hydrolysis performed using the elongation factor Tu.

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
