# Peer review of "The Reverse Side of a Coin: “Factor-Free” Ribosomal Protein Synthesis In Vitro is a Consequence of the In Vivo Proofreading Mechanism"

_biomolecules, 2019, doi:10.3390/biom9100588_

Round 1
Reviewer 1 Report
This manuscript describes the analysis of translation mechanism during decoding. The unique author of this manuscript is comparing so-called factor-free translation to in vivo protein biosynthesis. Based on experimental studies described in the literature, the author concludes that the in vivo proofreading mechanism is a mechanism to prevent miscoding issues from factor-free translation. Without any experimental data, the author says that his manuscript elucidates this issue.
The analysis presented in this manuscript is based on selected publications from the literature. Unfortunately, a big part of the literature is totally ignored thereby leading to speculations rather than conclusions. For instance, the detailed mechanistic studies from the Rodnina lab and others are not taken into account at all in this analysis. More precisely, the author speculates for instance on the fact that the binding of aa-tRNA to the ribosomal A-site is fully reversible in the factor-free pathway while being irreversible with EF-Tu due to GTP hydrolysis that makes it irreversible. This statement is not correct, in the case of near-cognate tRNAs, GTP hydrolysis is also observed like with cognate tRNA, still the near-cognate tRNA is released indicating that GTP hydrolysis is not always associated with irreversible binding. There are many publications reporting experiments demonstrating this effect that are totally ignored in this manuscript. Therefore, this reviewer does not recommend this manuscript for publication without any experimental data demonstrating the author’s conclusions and substantial manuscript remodelling integrating the missing literature.
Minor point:
- the figure 5 is somehow misleading because it does not integrate time, indeed as stated by the author the aa-tRNA binding with factor is binding faster that the aa-tRNA factor free, it would be informative to include the time scale in the x-axis.
Author Response
Author’s response to Reviewer 1:
I thank the Reviewer for reading my paper and I am grateful for his/her comments.
However:
1) I disagree with that part of the Reviewer’s phrase
> Based on experimental studies described in the literature, the author concludes that the in vivo proofreading
> mechanism is a mechanism to prevent miscoding issues from factor-free translation.
which is given above in bold. In fact, this phrase must read as
> prevent miscoding issues using a reaction pathway that can be also used by the factor-free translation in vitro.
This difference, or rather this Reviewer’s mistake, shows that she/he did not get the sense of my paper…
2) Further, the Reviewer did not understand that the essence of my paper is based on two fundamental facts only:
The essence of this paper is based on two fundamental facts: (i) that there are two pathways of aa-tRNA attachment to the ribosome (and thus, the reverse reactions will detach aa-tRNA from the ribosome), and (ii) that one of these pathways is GTP-dependent and is accompanied by a large free energy decrease due to GTP hydrolysis, while another pathway is not accompanied by hydrolysis and the corresponding free energy decrease.
These two facts are enough to conclude that, if the resulting (aa-tRNA)-ribosome complex is unstable, the aa-tRNA will detach from the ribosome using the reverse movement along just the GTP-independent pathway, because the usage of reverse movement along the GTP-dependent pathway is prohibited by the necessity of the large free energy increase. This is normal, strict and unavoidable physical logic (sometimes called “detailed balance principle”), which, I am afraid, the Reviewer does not understand and therefore requests some additional experimental evidences (which are not necessary at all), and accuses me that I ignore them.
Now the above considerations are included in the Conclusion of the paper.
3) At last, the Reviewer writes that I “speculate” that “GTP hydrolysis that makes it [binding of aa-tRNA to the ribosomal A-site] irreversible” and continues “This statement is not correct, in the case of near-cognate tRNAs, GTP hydrolysis is also observed like with cognate tRNA, still the near-cognate tRNA is released indicating that GTP hydrolysis is not always associated with irreversible binding.”
This is a clear misunderstanding. The GTP hydrolysis makes the aa-tRNA binding to the ribosomal A-site irreversible along the GTP-dependent pathway only, and the main point of my paper is that the aa-tRNA binding is reversible due to existence of another, the factor-free pathway; this is illustrated by Figure 6.
4) On the “Minor point”: I have modified Figure 5 and the corresponding legend so as to make it clearer.
Thanks to the Reviewer’s comments, I included a really important reference to the Rodnina’s paper (now Ref. [8]) and noted (in page 1) that the kinetic-based step of initial selection aa-tRNA is not addressed in my paper (which is concentrated on the next quality-control step, i.e., the proofreading itself). Also, I (with a help of professional translator) have done minor corrections of style spelling.
Reviewer 2 Report
Subsequently to the authors’ previous paper appeared in Molec. Biol. (Mosk) (the reference number 8), the author has reported in this manuscript a proofreading mechanism, which is carried out through a factor-free removal of Aa-tRNA, whose anticodon is not complementary to the ribosome-exhibited mRNA codon. This is an interesting manuscript. So, the manuscript could be accepted, after the revisions according to my suggestions and comments described below, if those are reasonable.
Minor revisions:
It is important to show in a scientific paper what is new. Therefore, first, author should explain differences between the contents described in this manuscript and in the previous paper, which has been published in Molec. Biol. (Mosk) (the reference number 8). It seems to me that the rejection of the non-complementary Aa-tRNA is carried out through one step process.If so, two blue arrow heads except one on the center of blue dashed two way arrow(s) should be removed from the reversible factor-free pathway in Figure 6 to align the blue dashed arrow(s) and the red solid arrow. In addition, green arrow head should be connected with the base of the black one way arrow showing the transpeptidation & translocation, similarly to the red solid arrow. Line 31: “As-tRNA” should be “Aa-tRNA”.Author Response
Author’s response to Reviewer 2:
I thank the Reviewer for attentive reading my paper and I am grateful for his/her comments.
However, I do not feel that it is appropriate to fulfill his/her request on explaining, in the text of this paper,
> differences between the contents described in this manuscript and in the previous paper.
This can mislead a reader who did not read the variant published in Molec. Biol. (Mosk). But I feel that it is useful to explain this difference to the Reviewer and the Editors.
The main difference is a new and much more extensive discussion of the found relationship of the factor-free ribosomal protein synthesis in vitro and the in vivo proofreading mechanism, which inscribes the proofreading in the context of ribosomal functioning. Also, in the new version of the manuscript I have separated the proofreading itself, that occurs after the irreversible GTP hydrolysis, when the Aa-tRNA is already somehow bound to the ribosome, and the initial kinetic selection of aa-tRNAs.
As a result, the list of references to literature has been extended.
All the figures in this manuscript are different from those used in Ref. [8] (now – [9]).
The resulting modifications of the revised manuscript can be summarized as follows: (i) very short description of differences between Ref. [8] (now – [9]) and current paper has been included, (ii) the excess arrowheads in Figure 6 have been removed, and (iv) the typo found by the Reviewer has been corrected.